# Multi-Omics Profiling in Marfan Syndrome: Further Insights into the Molecular Mechanisms Involved in Aortic Disease

**DOI:** 10.3390/ijms23010438

**Published:** 2021-12-31

**Authors:** Judith M. A. Verhagen, Joyce Burger, Jos A. Bekkers, Alexander T. den Dekker, Jan H. von der Thüsen, Marina Zajec, Hennie T. Brüggenwirth, Marianne L. T. van der Sterre, Myrthe van den Born, Theo M. Luider, Wilfred F. J. van IJcken, Marja W. Wessels, Jeroen Essers, Jolien W. Roos-Hesselink, Ingrid van der Pluijm, Ingrid M. B. H. van de Laar, Erwin Brosens

**Affiliations:** 1Department of Clinical Genetics, Erasmus MC, University Medical Center Rotterdam, 3015 GD Rotterdam, The Netherlands; j.m.a.verhagen@erasmusmc.nl (J.M.A.V.); burger.j.emc@gmail.com (J.B.); h.bruggenwirth@erasmusmc.nl (H.T.B.); m.vandersterre@erasmusmc.nl (M.L.T.v.d.S.); m.vandenborn@erasmusmc.nl (M.v.d.B.); m.w.wessels@erasmusmc.nl (M.W.W.); i.vandelaar@erasmusmc.nl (I.M.B.H.v.d.L.); 2Department of Molecular Genetics, Erasmus MC, University Medical Center Rotterdam, 3015 GD Rotterdam, The Netherlands; j.essers@erasmusmc.nl; 3Department of Cardiothoracic Surgery, Erasmus MC, University Medical Center Rotterdam, 3015 GD Rotterdam, The Netherlands; j.a.bekkers@erasmusmc.nl; 4Center for Biomics, Erasmus MC, University Medical Center Rotterdam, 3015 GD Rotterdam, The Netherlands; a.dendekker@erasmusmc.nl (A.T.d.D.); w.vanijcken@erasmusmc.nl (W.F.J.v.I.); 5Department of Pathology and Clinical Bioinformatics, Erasmus MC, University Medical Center Rotterdam, 3015 GD Rotterdam, The Netherlands; j.vonderthusen@erasmusmc.nl; 6Department of Clinical Chemistry, Erasmus MC, University Medical Center Rotterdam, 3015 GD Rotterdam, The Netherlands; marinazajec@hotmail.com; 7Department of Neurology, Erasmus MC, University Medical Center Rotterdam, 3015 GD Rotterdam, The Netherlands; t.luider@erasmusmc.nl; 8Department of Vascular Surgery, Erasmus MC, University Medical Center Rotterdam, 3015 GD Rotterdam, The Netherlands; 9Department of Radiation Oncology, Erasmus MC, University Medical Center Rotterdam, 3015 GD Rotterdam, The Netherlands; 10Department of Cardiology, Erasmus MC, University Medical Center Rotterdam, 3015 GD Rotterdam, The Netherlands; j.roos@erasmusmc.nl

**Keywords:** thoracic aortic aneurysms, Marfan syndrome, mitochondria, proteomics, RNA-seq

## Abstract

Thoracic aortic aneurysm is a potentially life-threatening disease with a strong genetic contribution. Despite identification of multiple genes involved in aneurysm formation, little is known about the specific underlying mechanisms that drive the pathological changes in the aortic wall. The aim of our study was to unravel the molecular mechanisms underlying aneurysm formation in Marfan syndrome (MFS). We collected aortic wall samples from *FBN1* variant-positive MFS patients (n = 6) and healthy donor hearts (n = 5). Messenger RNA (mRNA) expression levels were measured by RNA sequencing and compared between MFS patients and controls, and between haploinsufficient (HI) and dominant negative (DN) *FBN1* variants. Immunohistochemical staining, proteomics and cellular respiration experiments were used to confirm our findings. *FBN1* mRNA expression levels were highly variable in MFS patients and did not significantly differ from controls. Moreover, we did not identify a distinctive TGF-β gene expression signature in MFS patients. On the contrary, differential gene and protein expression analysis, as well as vascular smooth muscle cell respiration measurements, pointed toward inflammation and mitochondrial dysfunction. Our findings confirm that inflammatory and mitochondrial pathways play important roles in the pathophysiological processes underlying MFS-related aortic disease, providing new therapeutic options.

## 1. Introduction

Thoracic aortic aneurysms most commonly involve the aortic root and/or proximal ascending aorta and are often discovered incidentally. However, if left untreated, thoracic aortic aneurysms can result in aortic dissection, rupture or sudden death. Approximately 20% of individuals with thoracic aortic aneurysms and dissections have affected family members, suggesting a genetic predisposition [1]. This is also referred to as heritable thoracic aortic disease (HTAD) [2].

A major breakthrough in our understanding of HTAD came in 1991, when Dietz et al. identified *FBN1* (MIM 134797) encoding the extracellular matrix protein fibrillin-1, as the causal gene for Marfan syndrome (MFS (MIM 154700)) [3]. Patients with MFS typically present with aortic root dilatation, ectopia lentis and skeletal features. A pathogenic variant in *FBN1* can be identified in over 90% of patients that fulfill the Ghent criteria for MFS [4,5]. Though MFS was initially regarded as a structural disorder of the extracellular matrix caused by the deficiency of fibrillin-1, abnormal signaling of the transforming growth factor-beta (TGF-β) pathway has emerged as the main pathophysiological mechanism. Nonetheless, the exact pathophysiological processes underlying aortic aneurysms in MFS remain unclear.

Transcriptome profiling can help to elucidate the functional consequences of disease-associated gene variants. Characterization of affected biological pathways and/or gene networks may enhance the development of more effective therapies for thoracic aortic disease in MFS, and possibly, other genetic aortopathies. To our knowledge, transcriptome profiling has been described in only one patient with MFS [6]. Using single-cell RNA sequencing, the authors identified novel markers of smooth muscle cell phenotype modulation during aortic aneurysm development, including enhanced TGF-β signaling and Kruppel-like factor 4 (*Klf4*) overexpression, confirming their observations of the *Fbn1^C1041G/+^* MFS mouse model. Recently, Oller et al. performed transcriptome profiling in *Fbn1^C1039G/+^* mice to study the role of mitochondrial dysfunction in MFS, and confirmed their findings in MFS patients [7]. Here, we applied RNA-seq in full-thickness aortic biopsies from two groups of patients with MFS, with either haploinsufficient (n = 3) or dominant negative (n = 3) *FBN1* variants, to determine the molecular mechanisms affected by these two variant types.

## 2. Results

### 2.1. Patient Characteristics

The mean age at ascending aortic aneurysm surgery was not significantly different between MFS patients (30.2 ± 11.8 years) and controls (37.8 ± 18.1 years, *p* = 0.42). Four out of six MFS patients were male (67%) compared to two out of five controls (40%, *p* = 0.57). The maximum aortic diameter in MFS patients averaged 51.5 ± 4.0 mm (corresponding z-scores 7.03 ± 2.34). All MFS patients and controls had normal tricuspid aortic valves. The clinical and genetic features of MFS patients are presented in Table 1.

### 2.2. Histopathological Findings

Aortic wall specimens were available for histologic examination from five out of six MFS patients. All specimens had histopathologic features of moderate to severe overall medial degeneration (Appendix A), i.e., disorganization and fragmentation of elastic fibers, smooth muscle cell nuclei loss, and mucoid extracellular matrix accumulation [8,9]. No clear signs of inflammation were noted.

### 2.3. Gene Expression Profile in Healthy Aortas

We generated more than 34 million reads per sample (mean 43.5 ± 15.5 million reads). Of these reads, 92% to 99% aligned to the human reference genome. A total of 10,096 transcripts showed detectable expression (defined as mean expression of ≥2 reads per kilobase per million) in the ascending aorta of the control group. Twenty-six transcripts were highly expressed (≥1000 reads per kilobase per million) (Appendix A), including 13 out of the 14 mitochondrial protein-coding genes and the known HTAD genes *ACTA2* (aortic smooth muscle actin (MIM 102620)) [10] and *BGN* (biglycan (MIM 301870)) [11]. The list further includes several genes that code for calcium-binding and cytoskeletal proteins, such as *MYL9* (myosin light chain 9 (MIM 609905)), *TAGLN* (transgelin (MIM 600818)), *S100A6* (S100 calcium binding protein A6 (MIM 114110)) and *TMSB4X* (thymosin beta 4 X-linked (MIM 300159)), that represent good candidate genes for HTAD. Of note, we did not find rare, potentially deleterious variants in these genes in our in-house cohort of 179 patients with unexplained thoracic aortic disease. Principal component analysis (PCA) of gene expression data in CLC Genomics Workbench (QIAGEN) showed separated clusters of MFS samples and controls (Figure 1A). Hierarchical clustering analysis showed differences in gene expression based on the predicted functional effect of the *FBN1* variants: the three DN samples cluster together as well as two of the three HI samples. However, one HI sample was clearly different from the other samples (Figure 1B). In addition, PCA did not indicate a gender-dependent effect (Appendix A).

### 2.4. Differential Expression and Pathway Analysis

The DESeq2 algorithm was used to detect differentially expressed genes in a pairwise comparison between MFS and controls. Differentially expressed genes were uploaded into IPA for further analysis.

#### 2.4.1. FBN1 Expression Is Highly Variable and Not Significantly Different from Healthy Aortas

The mean *FBN1* mRNA levels in aortic tissue from all patients with MFS did not significantly differ from controls (*p* = 0.2963). We then subdivided the MFS patients into those with predicted haploinsufficient (HI, n = 3) and those with dominant negative (DN, n = 3) *FBN1* variants (Table 1). Gene expression profiles of these subgroups did not differ substantially. *FBN1* mRNA expression levels were not significantly (*p* = 0.1178) lower in MFS patients with HI variants (mean 360 ± 74 counts per million) compared to MFS patients with DN variants (mean 772 ± 351 counts per million) (Table 1). However, the trend suggests that our in silico prediction of the functional effect of the *FBN1* variants might be correct. Inter-individual variability was high, with *FBN1* mRNA levels ranging from 38% (15q21 deletion) to 150% (homozygous non-cysteine missense variant) of the mean expression level in controls. In general, fibrillin-1 mRNA and protein levels showed poor correlation (Table 1), suggesting different regulatory mechanisms. The expression of other TGF-β pathway genes was not significantly different between the HI and DN group, but again we observed large inter-individual variation (Figure 2A).

#### 2.4.2. Overrepresentation of Inflammatory and Mitochondrial Pathways in MFS Patients

The DESeq2 algorithm yielded a total of 1548 differentially expressed genes between MFS samples and controls: 743 genes were upregulated and 805 were downregulated (Appendix A). The top 10 up- and downregulated genes, excluding pseudogenes and undefined genes, are displayed in Table 2. Of note, approximately half of these genes are involved in the processes of transcription and translation. Pathway analysis identified 62 canonical pathways significantly enriched by genes that were differentially expressed between MFS and controls (Appendix A), including 16 pathways involved in inflammation and 8 pathways indicating mitochondrial dysfunction. The top 10 enriched pathways are displayed in Figure 3A. Surprisingly, neither the canonical TGF-β signaling pathway nor pathways involved in actin cytoskeleton organization and cell adhesion and migration were among the significantly enriched pathways. Targeted assessment of the canonical TGF-β signaling pathway revealed significantly increased expression of the ligand TGFB2 (fold change 2.3, FDR adjusted *p*-value 0.03). The expression of the TGF-β receptors and intracellular signaling molecules was not significantly altered. Top-ranked network functions identified by IPA were associated with lipid metabolism and inflammatory response. Next, we analyzed TGF-β pathway gene expression in individual MFS patients. The results are depicted in Figure 2A. Overall, TGF-β signaling was not significantly altered. However, the expression of several individual genes (i.e., *FBN2* (MIM 612570), *MMP9* (MIM 120361) and *TGFB2* (MIM 190220)) was highly variable among the patients. Upstream regulator analysis predicts upregulation of factors involved in transcription regulation and DNA methylation. Regulators predicted to be downregulated include genes involved in immune response (Figure 3B).

### 2.5. Immunohistochemistry

We studied the expression of several members of the TGF-β pathway, including TGF-β1, phosphorylated SMAD2 (pSMAD2), and connective tissue growth factor (CTGF), by immunohistochemistry in the wall of the proximal ascending aorta of MFS patients and controls. Labeling intensity and distribution of all these markers were comparable between these groups (Figure 2B). In both groups, CTGF and TGF-β1 showed universally strong staining with a cytoplasmic and nuclear pattern in medial VSMCs, while pSMAD2 showed variably strong but diffusely positive nuclear staining in VSMCs of the media. This confirms the apparent lack of a TGF-β signature in our MFS samples.

### 2.6. Proteomics Analysis

We next performed proteomics on aortic wall samples from MFS patients and healthy controls. Pathway analysis was performed on a selection of proteins that were either present or absent in MFS patient samples compared to controls, designated over- and underrepresented, respectively. We identified 169 proteins that were overrepresented and 309 proteins that were underrepresented in patient samples compared to controls. Of the 478 deregulated proteins, 475 were successfully mapped in IPA. We performed both a pathway and an upstream regulator analysis to examine changes in the processes involved. Pathway analysis pointed towards mitochondrial dysfunction and oxidative phosphorylation as the most significantly affected pathways (Figure 4A), thereby strengthening the RNA-seq results (Figure 3A). In addition, pathways involved in cytoskeleton integrity and cellular signaling are differentially regulated (Figure 4A). The IPA Upstream Regulator Analysis predicts upregulation of factors involved in cell cycle control and DNA methylation. Regulators predicted to be downregulated comprise genes involved in mitochondrial respiration and metabolism, including *PPARGC1A* (PPARG coactivator 1 alpha (MIM 604517)), *PPARA* (peroxisome proliferator activated receptor alpha (MIM 170998)) and *PPARD* (peroxisome proliferator activated receptor delta (MIM 600409)) (Figure 4B). Hence, the proteomics data, in addition to the RNA-seq results, indicate possible alterations in mitochondrial function. The mitochondrial dysfunction network representation (Figure 4C) would predict that mitochondrial respiration is impaired, as all proteins within this network are underrepresented, which we therefore investigated further.

### 2.7. Mitochondrial Respiration

Mitochondrial respiration was determined in cultured VSMCs from our patients with haploinsufficient (HI) and dominant negative (DN) *FBN1* variants, except for the patient with the homozygous c.7003C>T variant (no cultured VSMCs available). In VSMCs with a HI variant, we observed a significantly decreased basal OCR compared to control VSMCs (Figure 5A,C, *p* < 0.0001). When treated with FCCP, VSMCs with a HI variant presented with a decreased maximum OCR compared to controls (Figure 5A,D, *p* < 0.01, *p* < 0.0001). VSMCs with a DN variant showed a similar basal OCR compared to controls (Figure 5B,C). Adding FCCP led to an increased maximum OCR compared to controls in VSMCs harboring the DN variant c.3506G>T, while it did not in VSMCs with the DN variant c.4954T>C (Figure 5B,D, *p* < 0.01). Taken together, our data show that VSMCs with a HI *FBN1* variant present with a decreased basal and maximum OCR, while VSMCs with a DN *FBN1* variant do not.

## 3. Discussion

In aortic samples from MFS patients, we identified a gene expression signature indicative of inflammation and mitochondrial dysfunction. Inflammation is a substantial contributor to atherosclerotic vascular disease, such as coronary artery disease [12]. In addition, chronic inflammation of the aortic wall is a typical histological feature of aneurysms located in the abdominal aorta [13]. Recent studies have demonstrated that inflammation is also involved in the pathogenesis of thoracic aortic disease, irrespective of a genetic predisposition [14,15,16]. Though initially described as a non-inflammatory condition, these studies suggest that inflammatory cells, cytokines and metalloproteinases contribute to medial degeneration in both familial and sporadic thoracic aortic disease.

Mitochondrial dysfunction, leading to overproduction of reactive oxygen species (ROS), reduced ATP production, and activation of cell death programs, has also been established as an important factor in various cardiovascular diseases [17]. The role of mitochondrial dysfunction in thoracic aortic disease, however, is less well studied. In VSMCs, ROS have been linked to many physiological processes such as growth, differentiation and migration [18]. Impaired mitochondrial function, reflected by decreased oxygen consumption and increased acidification rates in aneurysmal aortas from *Fibulin-4^R/R^* mice was recently described by van der Pluijm et al. [19]. Furthermore, their experiments demonstrated decreased oxygen consumption rates in VSMCs from *Tgfbr1^M318R/+^* mice, a mouse model for Loeys–Dietz syndrome (LDS), and skin fibroblasts from patients with MFS and LDS. In a recent study by Oller et al. [7], in which transcriptomic and metabolic analysis was performed using aortas from a *Fbn1^C1039G/+^* murine MFS model and MFS patients, mitochondrial dysfunction, together with mitochondrial DNA depletion, was identified as a driving force in the pathogenesis of aortic disease. Excitingly, treatment of *Fbn1^C1039G/+^* mice with nicotinamide riboside, a NAD^+^ precursor that enhances mitochondrial metabolism by increasing *Pgc1a* and *Tfam* expression, was able to revert the development of aortic aneurysm and to restore histological features of medial degeneration. Our current findings support the concept that altered mitochondrial function plays a role in the pathogenesis of aortic disease in MFS. Interestingly, mitochondrial respiration experiments showed that VSMCs with a HI *FBN1* variant exhibited decreased oxygen consumption under basal conditions and after stimulation, whereas VSMCs with a DN *FBN1* variant did not (Figure 5C,D). These data suggest that mitochondrial respiration, which reflects the activity of the electron transport chain complexes, is only affected by *FBN1* haploinsufficiency. However, considering the limited number of samples and variants tested, we cannot exclude an effect of DN *FBN1* variants on mitochondrial respiration, and further confirmation of our findings is needed.

We observed high inter-individual variability in *FBN1* mRNA expression levels in aortic tissue from MFS patients (Table 1). In patients with *FBN1* variants that are predicted to lead to haploinsuffiency, mutant *FBN1* transcript was indeed absent or present at a very low level (0–6%), therefore excluding large variations in nonsense-mediated mRNA decay as a cause of the variable expression in these patients. As suggested previously, variable expression of the normal allele likely contributes to the variability in *FBN1* mRNA expression, and may depend on functional polymorphisms in the promotor region or trans-acting regulators [20,21].

The top differentially expressed genes among MFS samples and controls encompassed several extracellular matrix genes, including *OGN* (osteoglycin (MIM 602383)), *ASPN* (asporin (MIM 608135)), *ECM2* (extracellular matrix protein 2 (MIM 603479)) and *FGFR2* (fibroblast growth factor receptor 2 (MIM 176943)). This might result from disorganization of extracellular matrix due to dysfunctional fibrillin-1, or reduced aortic wall strength due to reduced levels of normal fibrillin-1 [22]. We did not find a distinctive TGF-β gene expression signature in MFS patients. This might be related to the advanced stage of disease, i.e., severe aortic dilatation requiring surgical intervention. Though early studies reported increased activation of the TGF-β signaling pathway as the key mechanism in aneurysm formation, more recent data from mouse models and human genetic studies reveal a dual role of TGF-β signaling in aneurysm formation, depending on, e.g., the developmental origin of VSMCs, and suggest that additional mechanisms and pathways may be involved, such as the renin–angiotensin system and cytoskeleton dynamics [23,24,25]. Alternatively, by using full-thickness aortic wall samples we obtained results from a mixed population of cells, which could have masked the role of TGF-β signaling. Therefore, the use of VSMCs alone could shed further light on potentially important cellular processes involved in aneurysm formation. Proteomics and immunohistochemical analysis of various components of the TGF-β pathway did not reveal any differences in protein expression either. Altogether, our findings suggest that targeting the TGF-β signaling pathway might not be an effective treatment strategy at this stage of disease.

Gene expression profiles may alter with disease progression and medical treatment [26]. As a consequence, we likely missed relevant changes in gene expression that occurred early in the disease process. Our dataset does, however, provide important insight into the molecular mechanisms involved in disease progression. This knowledge may help to develop new therapeutic approaches to delay or prevent vascular complications and/or the need for surgical intervention. In fact, anti-inflammatory drugs may be effective in reducing aneurysm progression. Interestingly, 3-hydroxy-3-methylglutaryl-coenzyme A reductase inhibitors, also known as statins, which are widely prescribed for their lipid-lowering effect, also possess anti-inflammatory properties [27]. A systematic review and meta-analysis involving more than 80,000 patients with abdominal aortic aneurysms (AAA) recently confirmed that statin therapy is associated with reduced AAA growth, rupture rate and perioperative mortality of elective surgical repair [28]. Ex vivo experiments in human AAA tissues suggest that statins inhibit the Rac1/NF-κB pathway, with subsequent suppression of matrix metalloproteinase (MMP)-9 and cytokine secretion [29]. To date, only two studies have investigated the effect of statins in thoracic aortic aneurysms [30,31]. Both studies suggested better outcomes (e.g., improved survival free from dissection and rupture or increased interval to surgery) among patients receiving statins. Our data support the idea that anti-inflammatory therapy should be investigated further as an additive treatment in patients with thoracic aortic disease. Future studies should also focus on possible serologic markers for inflammation to help guide treatment decisions.

Restoring mitochondrial function may also be an effective strategy to prevent or slow progression of aneurysm formation in MFS. Therapeutic approaches in inherited mitochondrial disorders typically focus on maintaining optimal health, by preventing worsening of symptoms during illness and physiological stress (e.g., dehydration and prolonged fasting) and avoiding mitochondrial toxins. However, scientific evidence that current interventions, such as the use of vitamins supplements and exercise therapy, do alleviate mitochondrial disease manifestations is limited [32,33]. Based on our results and findings from previous studies, pharmacological intervention aimed at increasing the expression of peroxisome proliferator-activated receptor gamma coactivator 1-alpha (PPARGC1A or PGC1A) might be an appropriate therapeutic strategy for managing MFS-related aortic disease. PGC1A is a transcriptional coactivator that acts as a major regulator of mitochondrial biogenesis and function. Previous studies have shown that epoxyeicosatrienoic acid effectively induces PGC1A-mediated downstream signaling [34]. However, upstream regulation of PGC1A using nicotinamide riboside (or other NAD^+^ precursor supplements) may be even more successful, considering the recent findings in *Fbn1^C1039G/+^* mice [7]. Upstream regulator analysis further revealed that metabolic pathways involved in carbohydrate and lipid metabolism may serve as therapeutic targets, again suggesting the potential benefit of diet and exercise interventions as well as the use of statins. Further studies should therefore focus on metabolomic analysis in MFS patients, to further explore these theories and gain more insight into the potential therapeutic targets. Because the mitochondrial dysfunction observed in our aortic samples is probably a localized abnormality related to disease progression, we do not expect that MFS patients are at increased risk for serious perioperative complications as observed in patients with primary mitochondrial disease. In fact, elective surgical replacement of the proximal aorta in MFS patients is associated with very low morbidity and mortality [35]. However, like patients with primary mitochondrial disease, MFS patients may benefit from particular considerations in the perioperative period, including the selection and dosing of anesthetic agents and continuous temperature monitoring, to minimize the metabolic stress of surgery and prevent further damage to the blood vessels [36].

The limited number of available samples and large inter-sample variation might have masked the detection of other relevant biological processes. Due to limited resources, we did not include technical replicates in our experimental design. Therefore, we could not control for technical variation. Second, because we made use of surgically removed aortic tissue samples (indicating advanced stage of disease), we were not able to monitor changes in gene expression at different stages of disease progression. The same holds true for the immunohistochemistry and proteomics analysis. Third, RNA library preparation and sequencing procedures include a number of steps that might introduce biases in the resulting data [37]. The TruSeq Stranded mRNA Library Prep kit, for example, uses poly(A) selection for mRNA enrichment in order to remove abundant, unwanted ribosomal RNA, and thereby improving the sequencing depth of mRNA. However, this procedure also removes other non-polyadenylated transcripts, including multiple histone-encoding transcripts, which can therefore not be studied in downstream analysis.

In summary, both inflammation and mitochondrial dysfunction seem to play important roles in the pathogenesis of MFS. These pathways should be further explored as potential therapeutic targets for preventing aneurysm progression. Our data suggest that targeting the TGF-β signaling pathway might not be effective at advanced stages of thoracic aortic disease.

## 4. Materials and Methods

### 4.1. Study Population

Patients with molecularly proven MFS (i.e., harboring a pathogenic or likely pathogenic variant in the *FBN1* gene, see Appendix A, Methods) undergoing elective ascending aortic replacement at the Erasmus University Medical Center between January 2015 and May 2018 were included in this study. All patients underwent structured clinical evaluation including medical and family history, physical examination, transthoracic echocardiography and preoperative computed tomography angiography of the thoracic aorta. Aortic diameters were determined at the level of the aortic annulus, sinuses of Valsalva, sinotubular junction, proximal and distal ascending aorta, aortic arch, and proximal and distal descending aorta. Aortic valve morphology and function were documented at surgery. Variants in *FBN1* were classified as haploinsufficient (HI; production of only non-mutant fibrillin-1 protein) or dominant negative (DN; interference of mutant with non-mutant fibrillin-1 protein) using Alamut Visual (Interactive Biosoftware, Rouen, France), as described previously [38].

As controls, we collected residual aortic tissue from healthy donor hearts (HTx). The donor hearts, without preexisting cardiac disease and with normal aortic diameters, were stored on ice before transplantation. Excess ascending aortic tissue was removed before completing the anastomosis onto the recipient aorta. 

### 4.2. Sample Collection and Selection

Full-thickness aortic wall samples were snap-frozen in liquid nitrogen after surgical removal from either patient or donor heart, and stored at −80 °C until further processing. Samples were derived from the anterior side of the proximal ascending aorta (6 MFS and 5 HTx). A small piece of aortic tissue was used to isolate and culture vascular smooth muscle cells (VSMCs, see Appendix A, Methods). These VSMCs were used in the cellular respiration experiments. Statistical analyses were performed to compare the sample groups for significant differences in baseline characteristics. Data were expressed as mean ± standard deviation for continuous variables, and absolute numbers (percentage) for categorical variables. Comparisons were made using the Student’s *t*-test and Fisher’s exact test, respectively. All statistical tests were two-sided. Differences between groups were considered statistically significant at *p*-values < 0.05. Analyses were performed using Microsoft Excel or GraphPad Software.

### 4.3. Histology and Immunohistochemistry

Available surgical aortic samples from 5 out of 6 MFS patients were systematically examined for histopathologic findings of inflammatory and non-inflammatory aortic diseases according to current guidelines [8,39]. No residual aortic tissue was available for histopathological examination from one MFS patient included in the RNA sequencing analysis. All samples were simultaneously handled and examined after hematoxylin and eosin, Elastica van Gieson (elastin), Picrosirius red (collagen), and Alcian blue (proteoglycans) staining using standard techniques.

For immunohistochemistry, sections were deparaffinized and rehydrated, followed by fully automated antigen retrieval and immunostaining on a Roche Ventana Benchmark Ultra platform (Ventana Medical Systems, Tucson, AZ, USA). Primary antibodies used were: TGF-β1 (Abcam, ab53169), CTGF (Abcam, ab5097) and phospho-SMAD2 (Cell Signaling Technology, #3108). Negative controls were obtained by omitting the primary antibody.

### 4.4. RNA Sample Preparation and Sequencing

Frozen tissue samples were ground using a pre-chilled mortar and pestle, and subsequently disrupted and homogenized using a bead mill. Total RNA was extracted with the RNeasy Fibrous Tissue Mini Kit (QIAGEN). Residual genomic DNA was removed using the RNase-Free DNase Set (QIAGEN). RNA concentration, purity and integrity were assessed on the 2100 Bioanalyzer using the RNA 6000 Nano Kit (Agilent Technologies, Santa Clara, CA, USA). Only samples with an RNA integrity number (RIN) ≥7 were eligible for RNA-seq. Libraries were prepared using the TruSeq Stranded mRNA Library Prep Kit (Illumina, San Diego, CA, USA). The resulting DNA libraries were sequenced according to the Illumina TruSeq Rapid v2 protocol on an Illumina HiSeq2500 machine. Paired-end reads were generated of 100 base pairs in length. All relevant transcriptome data are within the manuscript and the Appendix A. Our ethics committee does not allow sharing of individual patient or control genotype information in the public domain. 

### 4.5. Differential Expression Analysis

RNA-seq reads were aligned to the human reference genome GRCh37 release 75 using HISAT2 version 2.0.4. For the alignment, the reference sequence was enhanced with exon junction sequences and single nucleotide polymorphisms. Aligned reads were sorted by read name; only uniquely aligned reads were counted per gene using htseq-count version 0.6.1 [40]. Normalization and differential expression analysis were performed using DESeq2 version 2.11.40.2 [41]. Read counts were normalized for transcript length and total number of mapped reads (RPKM) [42]. Genes with a fold change (FC) ≥1.5 and a false discovery rate (FDR) adjusted *p*-value < 0.05 were considered differentially expressed. Data were visualized using the Integrative Genomics Viewer (IGV).

Differentially expressed genes were uploaded into the Ingenuity Pathway Analysis (IPA) software, Summer Release 2018 (QIAGEN Bioinformatics). We performed a Core Analysis for each group, using default parameters, to predict the effects on pathways and biological functions, and discover plausible upstream regulators and mechanistic networks [43]. Pathways with a negative log10(*p*-value) greater than 1.3 were considered significant.

### 4.6. Liquid Chromatography-Mass Spectrometry Proteomics Measurements

Sample preparation and liquid chromatography (LC) measurements were performed as described previously [44]. Briefly, formalin-fixed paraffin-embedded aorta tissue samples of the same MFS patients were deparaffinized and rehydrated. Formalin cross-links were removed and samples were digested with trypsin overnight. LC was carried out on an UltiMate RSLCnano system (Thermo Fisher Scientific, Waltham, MA, USA). Shotgun mass spectrometry (MS) was performed on an Orbitrap Fusion Lumos mass spectrometer (Thermo Fisher Scientific, Waltham, MA, USA). High-resolution full-scan MS data were obtained using the following parameters: resolution of 120,000, AGC target of 4e5, maximum injection time of 50 ms and mass range of 375.00–1500.00 m/z. MS/MS spectra were obtained by HCD fragmentation applying 30% normalized collision energy. MS/MS was performed with a resolution of 30,000, AGC target of 1e4, maximum injection time of 50 ms and quadrupole isolation width of 1.6 m/z. Precursor ions that were selected once for MS/MS analysis were excluded for a duration of 60 s. 

### 4.7. Protein Identification

MS/MS spectra from the raw data files of each sample were converted into mgf files using ProteoWizard version 3.0 and used to carry out searches using Mascot version 2.3.02 against the UniProt database (selected for Homo sapiens, downloaded 15 November 2015, 20,194 entries). The enzyme specificity was set to trypsin and a maximum of four missed cleavages was allowed. Cysteine carbamidomethylation (+57 Da) was set as a fixed modification, and methionine oxidation (+16 Da) as a variable modification. The mass tolerance for precursor ions was 10 ppm, and the mass tolerance for fragment ions was 0.5 Da. Database search results were processed by Scaffold version 4.10.0 to merge the individual search results and filter peptide identifications. Protein confidence thresholds were set to a 1% false discovery rate (FDR), at least 2 peptides per protein identified, and a 1% FDR at the peptide level. FDRs were estimated by inclusion of a decoy database search generated by Mascot.

To perform IPA canonical pathway and upstream regulator analysis, all proteins identified in aortic wall samples from MFS patients and healthy controls were aligned and assessed for proteins only present in either one or both, as described previously [19]. Next, proteins only present in patient samples were designated ‘up’. Likewise, proteins only present in controls, and thus absent in patient samples, were designated ‘down’. For analysis, arbitrary signs of +5 and −5 were given to a protein designated present or absent, respectively. Upstream regulators were selected by bias-corrected z-score >2 and <−2 and a *p*-value < 0.01.

### 4.8. Cellular Respiration Experiments

Oxygen consumption rates (OCR) were measured using an XF-24 Extracellular Flux Analyzer (Seahorse Bioscience). Respiration was measured in XF assay media (non-buffered DMEM), in basal conditions and in response to 1 μM oligomycin (ATP synthase inhibitor), 1 μM fluoro-carbonyl cyanide phenylhydrazone (FCCP, mitochondrial oxidative phosphorylation uncoupler), 1 μM antimycin A (complex III inhibitor). VSMCs were seeded at a density of 30,000 cells/well and analyzed after 24 h. Optimal cell densities were determined experimentally to ensure a proportional response to FCCP with cell number. These experiments were performed with five independent wells per cell line for each time point, n = 2.

## Figures and Tables

**Figure 1 ijms-23-00438-f001:**
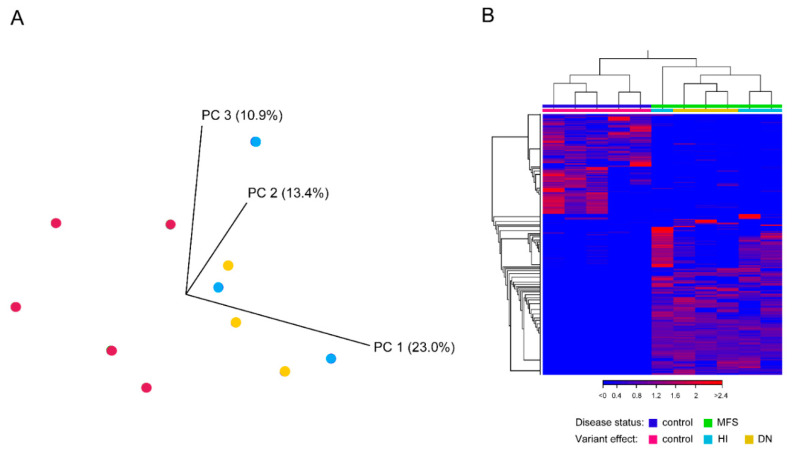
Visualization of gene expression data. (**A**) Three-dimensional scatter plots of principal component analysis on gene expression data. Cyan and yellow dots represent MFS samples with dominant negative (DN) and haploinsufficient (HI) *FBN1* variants, respectively. Magenta dots represent controls. The numbers in brackets correspond to the proportion of variance explained by the respective principal component. (**B**) Heat map generated from gene expression data showing hierarchical clustering of MFS samples related to the predicted effect of the *FBN1* variants.

**Figure 2 ijms-23-00438-f002:**
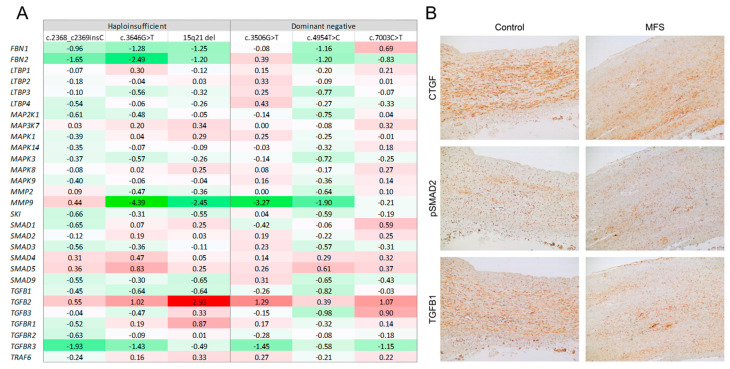
Targeted assessment of the canonical TGF-β signaling pathway. (**A**) Heatmap depicting TGF-β pathway deregulation scores (fold changes) in individual MFS patients. (**B**) Representative images of immunohistochemical analysis of TGF-β family proteins in aortic tissue of control and MFS patient. Labeling intensity and distribution of these markers were comparable between both groups.

**Figure 3 ijms-23-00438-f003:**
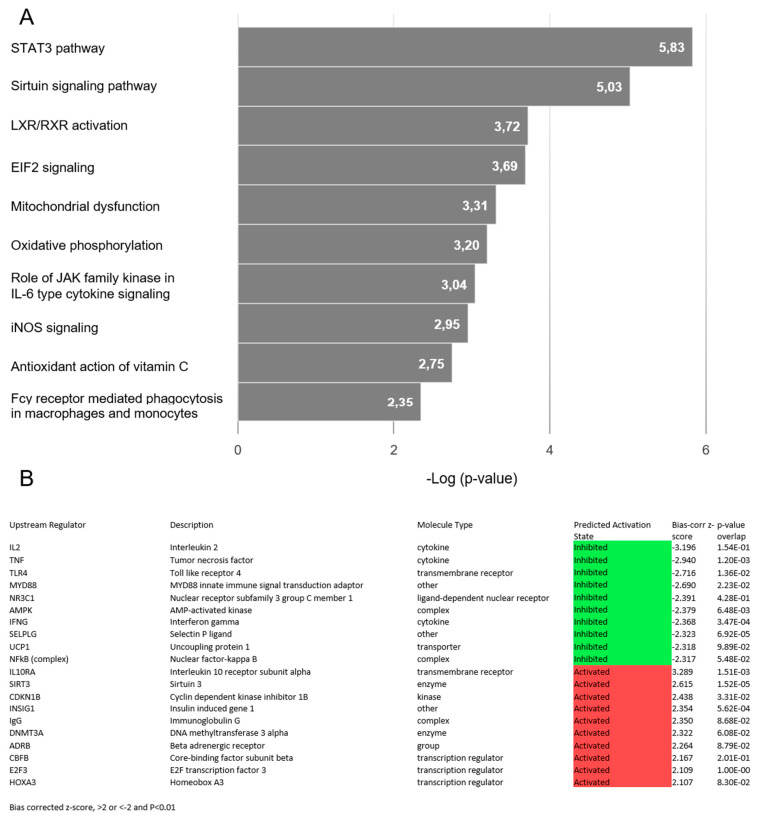
Expression analysis of RNA-seq data. (**A**) Graph depicting the top 10 canonical pathways (*p*-value < 0.01) derived from an IPA analysis based on differentially expressed genes (fold change ≥1.5 and false discovery rate adjusted *p*-value < 0.05) in aortic samples from MFS patients compared to controls. (**B**) Table depicting predicted activation or inhibition of key upstream transcriptional regulators (bias-corrected z-score >2 or <−2, *p*-value < 0.01) derived from an IPA analysis based on the observed gene expression changes.

**Figure 4 ijms-23-00438-f004:**
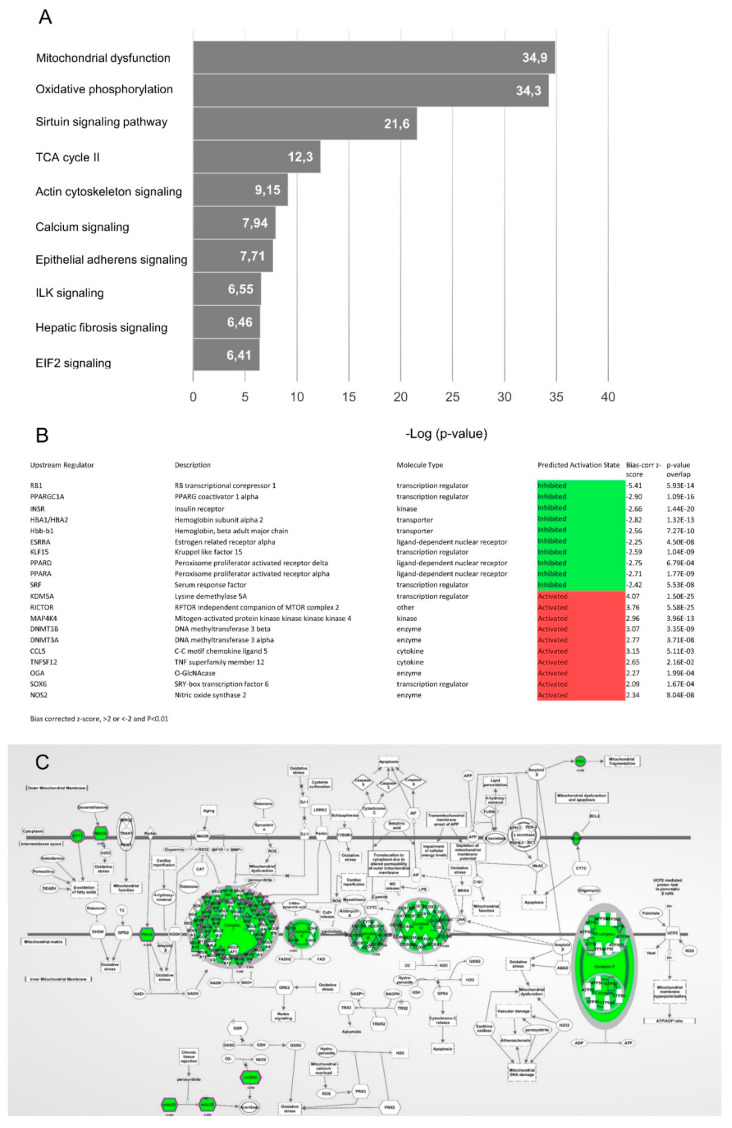
Expression analysis of proteomics data. (**A**) Graph depicting the top 10 canonical pathways (*p*-value < 0.01) derived from an IPA analysis based on differentially regulated proteins in aortic samples from MFS patients compared to controls. (**B**) Table depicting predicted activation or inhibition of key upstream transcriptional regulators (bias-corrected z-score >2 or <−2, *p*-value < 0.01) derived from an IPA analysis based on the observed protein expression changes. (**C**) Mitochondrial dysfunction network representation derived from IPA.

**Figure 5 ijms-23-00438-f005:**
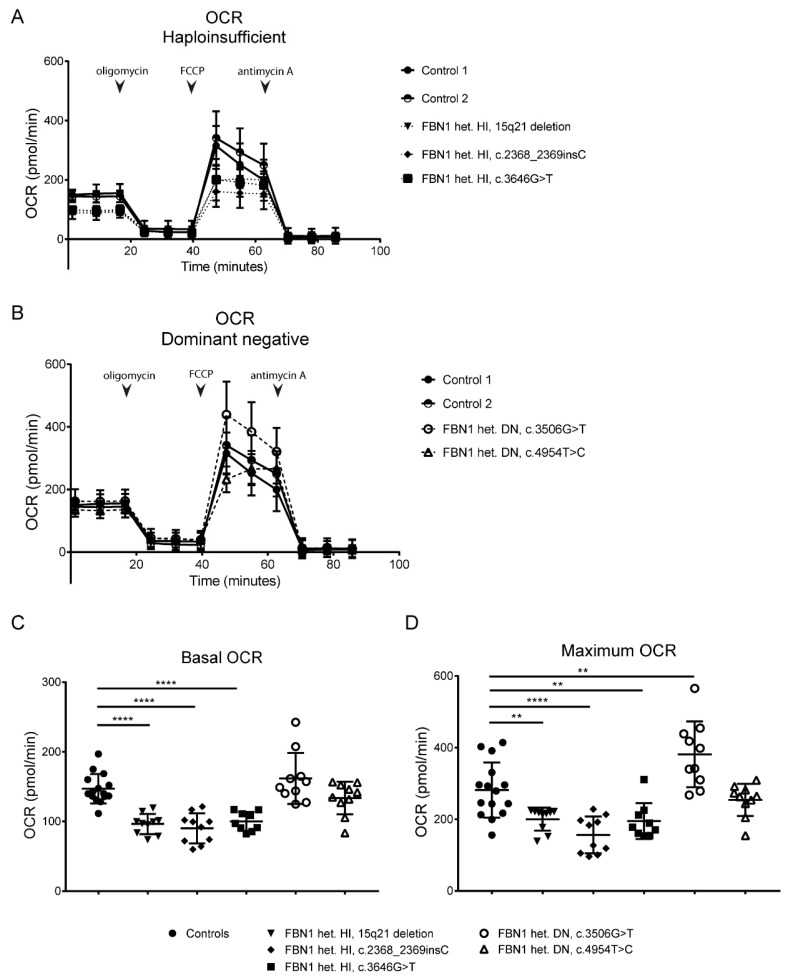
Decreased mitochondrial respiration in *FBN1* haploinsufficient VSMCs. (**A**) Oxygen consumption rate (OCR) of *FBN1* haploinsufficient (HI) VSMCs at basal level and in response to 1 μM oligomycin, 1 μM fluoro-carbonyl cyanide phenylhydrazone (FCCP) and 1 μM antimycin A. (**B**) Oxygen consumption rate (OCR) of *FBN1* dominant negative (DN) VSMCs at basal level and in response to 1 μM oligomycin, 1 μM fluoro-carbonyl cyanide phenylhydrazone (FCCP) and 1 μM antimycin A. Of note, data for controls are depicted twice, in both part A and B. (**C**) Basal OCR of *FBN1* mutated VSMCs. (**D**) Maximum OCR of *FBN1* mutated VSMCs after addition of 1 μM FCCP (** *p*-value < 0.01, **** *p*-value < 0.0001, Mann-Whitney U test). Results are expressed as mean ± SEM.

**Table 1 ijms-23-00438-t001:** Clinical and genetic features of patients with MFS.

**Part A: Clinical Features**						
**Age at operation (years)**	32	16	49	23	37	24
**Aortic diameter (mm)**	55	49	50	48	49	58
**Aortic z-score**	8.14	6.89	5.82	5.89	4.37	11.07
**Ectopia lentis**	+	+	−	+	+	+
**Systemic score**	8	4	7	5	8	8
**Part B: Genetic Features**						
**cDNA change**	c..2368_2369insC	c.3646G>T	15q21 deletion (including *FBN1*) ^‡^	c.3506G>T	c.4954T>C	c.7003C>T
**Protein change**	p.Cys790Serfs*12	p.Glu1216*		p.Gly1169Val	p.Cys1652Arg	p.Arg2335Trp
**Zygosity**	Heterozygous	Heterozygous	Heterozygous	Heterozygous	Heterozygous	Homozygous
**Predicted effect**	HI	HI	HI	DN	DN	DN
**Mutant allele**	0/141 (0%)	5/82 (6%)	Absent (0%)	78/199 (39%)	52/120 (43%)	400/400 (100%)
***FBN1* mRNA level ^†^**	57%	46%	38%	93%	59%	150%
**FBN1 protein level ^††^**	55%	NA	98%	106%	80%	96%

DN, dominant negative; HI, haploinsufficient; NA, not available. Reference sequence: NM_000138.4. ^†^ *FBN1* mRNA expression level compared with controls (n = 5). The mean level in controls was set at 100%. ^††^ FBN1 peptide count compared with controls (n = 6). The mean level in controls was set at 100%. ^‡^ 15q21.1q21.2 deletion of approximately 7.6 Mb encompassing *FBN1* and multiple other genes.

**Table 2 ijms-23-00438-t002:** Top 10 up- and downregulated genes in MFS compared with controls.

Symbol	Description	Location	FC	FDR
**Upregulated Genes**				
*MTRNR2L12*	MT-RNR2 like 12	Cytoplasm	18.3	1.94E-56
*LINC00965*	Long intergenic non-protein coding RNA 965	No data available	13.1	3.47E-18
*GOLGA8I*	Golgin A8 family, member I	Golgi apparatus	12.5	2.19E-17
*SCXA*	Scleraxis bHLH transcription factor	Nucleus	11.2	1.11E-15
*GOLGA8S*	Golgin A8 family, member S	Other	10.7	3.13E-15
*NBPF24*	NBPF member 24	Other	10.1	3.99E-33
*RASA4*	RAS p21 protein activator 4	Cytosplasm	8.1	2.57E-16
*ARL17A*	ADP Ribosylation Factor Like GTPase 17A	Golgi apparatus	6.9	1.19E-17
*ZNF84*	Zinc finger protein 84	Nucleus	6.8	1.26E-15
*PRND*	Prion like protein doppel	Plasma membrane	5.8	4.96E-07
**Downregulated Genes**				
*RPL18A*	Ribosomal protein L18a	Cytoplasm	−102.4	3.89E-210
*RPL21*	Ribosomal protein L21	Cytoplasm	−42.7	7.10E-280
*RPS26*	Ribosomal protein S26	Cytoplasm	−33.9	2.12E-103
*ADH1A*	Alcohol dehydrogenase 1A (class I), alpha polypeptide	Cytoplasm	−33.1	7.95E-52
*NAMPTL*	Nicotinamide phosphoribosyltransferase-like	No data available	−16.9	4.54E-20
*EIF3CL*	Eukaryotic translation initiation factor 3 subunit C like	Other	−14.2	1.06E-35
*F8A3*	Coagulation factor VIII associated 3	Nucleus	−7.1	2.24E-10
*H3-3A*	H3.3 histone A	Nucleus	−5.8	4.71E-28
*MAGED4B*	MAGE family member D4B	Other	−5.3	3.18E-06
*RPL9*	Ribosomal protein L9	Nucleus	−5.2	3.78E-07

FC, fold change; FDR, false discovery rate (Benjamini–Hochberg procedure); MFS, Marfan syndrome. Complete list of differentially expressed genes is provided in Appendix A.

## Data Availability

All relevant data are within the manuscript and the Appendix A. Our Ethics Committee does not allow sharing of individual patient or control genotype information in the public domain.

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
