# Peer review of "Multi-Omics Profiling in Marfan Syndrome: Further Insights into the Molecular Mechanisms Involved in Aortic Disease"

_ijms, 2021, doi:10.3390/ijms23010438_

Round 1
Reviewer 1 Report
This study provides valuable -omics information from aortic samples derived from several aortic samples of Marfan syndrome (MFS) patients, and complements the one available single cell RNA sequencing experiment of MFS aorta. However, a few limitations diminish the potential impact of the work:
1) In Figure 1A, it is not clear whether samples in the PCA segregate by disease status. From the figure, it appears that only samples from patients carrying dominant negative FBN1 mutations separate from controls in PCA1. Which transcripts are the major contributors to variance in PCA1, 2 and 3? Occasionally “outlier” artifactual transcripts can confound the PCA analysis and can be excluded from analysis.
2) An obvious limitation of bulk-RNA seq of the whole thickness of the aorta is confounding effect of various cell types, and “apple to oranges” comparisons when cellular composition varies between disease and controls. The authors try to compensate this limitation by “normalization” to smooth-muscle specific transcripts, in order to “deconvolute” bulk RNA-seq data. I am not convinced this is a computationally-appropriate approach and I suggest the authors consider using more appropriate methods such as those described here
http://bioconductor.org/packages/release/bioc/vignettes/granulator/inst/doc/granulator.html
Reviewer 2 Report
In this paper, the authors identify in aortic samples of MFS patients, a gene expression signature indicative of inflammation and mitochondrial dysfunction. The text is clear and results can be extremely interesting concerning clinical implications. However, some points should be better specify:
- The authors compared MFS versus healthy donor hearts (HTx) ascending aorta samples. The control group should be further characterized (how the aortic samples have been collected? which is the ascending aorta’s diameter of these patients?). The authors should better clarify also the main characteristics of the CABG group.
- In line 101, the authors state “available surgical aortic samples (from 5 out of 6 MFS patients that were used in the RNA sequencing analysis”. Why one patient was excluded? The authors should better explain in the text.
- In line 266, is the “in silico prediction” referred to the dominant-negative group?
- The authors show in Figure 3 Immuno-histochemical analysis of TGF-β family proteins in aortic tissue of control and MFS patient. The authors should better clarify in paragraph 3.5 the results obtained. In addition, authors in the footnotes of figure 3 describe sections A and B but relative references are missing in the figure.
- In Figure 4c, data concerning gene mutations associated with mitochondrial dysfunction should be better reported. Moreover, Figure 4c resolution should be better improved.
- The authors state that in aortic samples from MFS patients altered mitochondrial function plays a role in aneurysm formation. It has been previously described that the inflammatory infiltrate could correlate with the aortic dilatation, but not contribute to the disease progression. Is it possible to hypothesize the same mechanism for mitochondria dysfunction?
- From line 386 to 390, the authors postulate regarding the role of mitochondrial dysfunction presents in the aneurysm formation in MFS, especially the FBN1 haplo-insufficiency. Authors should better describe results obtained
- Authors could further speculate, into the discussion section, on possible applications of data obtained also in terms of surgical, beyond pharmacological, approach
Reviewer 3 Report
This manuscript reports transcriptome and proteome analyses to elucidate pathogenetic mechanisms in aortic aneurisms of Marfan patients. The major conclusions by the authors is that their data support previous reports that implicate inflammation and mitochondrial dysfunction in the pathogenesis. Although the study is based on a small number of samples, this is a study that addresses the questions in real patients, samples of whom are precious as they are very difficult to obtain. This is a very necessary evaluation of results from animal models and other studies using simplified model systems.
Aortic cell wall samples from 6 patients, 3 with likely haploinsufficiency and 3 with likely dominant negative pathology, and 5 controls from transplantation donor hearts and from patients that have undergone coronary artery bypass surgery are analysed by RNASeq and MS proteomics. In addition, cultured vascular smooth muscle cells from biopsies taken from the patients and controls are used for functional analysis of the mutation effects on mitochondrial respiration.
The manuscript has some serious shortcomings regarding the presentation of the results, description of methodology and it lacks sharpness in the arguments underpinning the conclusions. The arguments for mitochondrial dysfunction based on the OMICS analyses are quite speculative and the respiration analysis, which appears to be the best argument for an effect of haploinsufficiency mutations on mitochondrial respiration are not well underpinned and the description of these experiments is imprecise. In the following some specific points:
The 3d PCA graph representation in Figure 1A is difficult/impossible to appreciate. The same holds true for Supplemental Figure S2.
Lines 232,-235: Genes that represent good candidates for HTAD genes: What is the argument that highly transcribed genes should be good candidates to represent HTAD genes?
Lines 239,240: ‘Hierarchical clustering was not related to the predicted functional effect of the FBN1 variants (Figure 1B).’ The 3 DN samples cluster together and 2 of the 3 HI samples also…
3.6. L 293/294: Gene Ontology (GO) Enrichment Analysis revealed no statistically significant results. What were the criteria? This is quite surprising in light of the IPA analysis in Figure 2A that shows a series of significant pathways.
3.6.; lines 295-300 and methods 2.6. VSMC housekeeping gene correction: It appears suspicious to use the correction with expected housekeeping genes. The formula in 2.6. is imprecise; one needs to guess that the ratios for gene n in sample n to average of gene n in all samples are summed up for the 10 reference gene and the sum is then divided by n? Statistically the correction is questionable. DESEQ2 attempts to model the data so that differences between samples are minimized in order to identify the ‘real’ differences. Correcting this by a rather simple calculation appears not to be appropriate.
Figure 3A is described in the text before Figure 2. The heatmap with numbers is not easy to appreciate. A scale of the color code (fold changes?) would be better.
Figure 3: B shown above A; no label (A,B) in figure. Upper panel/ lower panel would be more appropriate.
Table 2 with top 10 up- and down-regulated genes contains 9 pseudogenes (of 20 genes differentially regulated!). Are pseudogene levels relevant?
Supplementary table S4: The connection of these pathways to mitochondria is unclear: iNOS Signaling, Antioxidant Action of Vitamin C, Production of Nitric Oxide and Reactive Oxygen Species in Macrophages, nNOS Signaling in Skeletal Muscle Cells. What is the connection to mitochondrial dysfunction? Mitochondrial Dysfunction comprises the genes of Oxidative Phosphorylation.
3.8. Proteomics analysis: How many proteins were detected? How were the samples normalized to each other? Proteins that were absent in one group and present in the other are given +5 or -5 respectively. This makes a bias to proteins that are lowly expressed and therefore only sampled in some of the MS runs. What were the average number of identified peptides for this subset in comparison to average number of identified peptides in the whole set?
Figure 4B: I do not see anything supporting the statement: ‘Regulators predicted to be downregulated include genes involved in mitochondrial respiration and metabolism.’ The basis how the Figure was made and what it actually depicts is unclear. Which genes are shown and is that based on their differential expression in the analysis? Was the cell count similar for control and mutant cells after 24 hours incubation?
Figure 4C is not readable.
3.9. Mitochondrial respiration: How were the The SEM error bars for the FCCP values are quite large. Which measuring point(s) were used for calculating the basal and maximum OCRs? The points are based on 5 wells for each sample (control, mutant) measured on 2 different days and all 10 values are shown. How big was the day to day variation (average of 5 wells on day 1 and day 2)?
Discussion
Lines 399, 400: ‘The top differentially expressed genes between MFS samples and controls encompassed several extracellular matrix genes.’ Which are these genes?
Lines 415, 416: ‘our findings suggest that targeting the TGF-β signaling pathway will not be an effective treatment strategy at this stage of disease.’ The data basis for this statement appears quite weak.

Round 2
Reviewer 1 Report
1) The color scale for hierarchical clustering in Fig 1B goes from 0 to +2.5-- I don't think that makes sense. The data should be centered so that the scale can be symmetric.
2) The rationale as to why HI, but not DN FBN1 mutations, would cause mitochondrial dysfunction, and how that would correlate with disease severity remains very weak. Given that both type of mutations cause disease, and mitochondrial dysfunction is only observed in one set, it seems that mitochondrial dysfunction is not really cause of disease at all. This should be strongly noted.
3) Figure 2A, we recommend authors use "." to indicate the decimal point, as opposed to ",". I know "," is used in some European countries but international readers may be confused.
4) Given that a more sophisticated de-convolution analysis was not possible, I think the authors should show data *not* normalized to VSMC housekeeping genes. They can also show "normalized" data if they wish to do so, but I remain skeptical this "normalization" is mathematically sound, given that disease itself can change expression of VSMC housekeeping genes in MFS sample.
5) The exact aortic location for shown histology should be noted (root/proximal ascending/distal ascending), as that may be relevant to specific alterations. Authors should also note that timing may be important and that late-stage findings may not reflect initiating events.
Reviewer 2 Report
The authors replied exhaustively to the clarifications requested. However, Figure 4c should be still improved concerning resolution.
Moreover, in the discussion section, a short review of mitochondrial dysfunction, according to literature, in terms of surgical approach on aortic disease should be presented, to speculate possible new perspectives.
Reviewer 3 Report
A number of points have been addressed, but I have still some concerns. I still hold that the mitochondrial dysfunction evidence is quite circumstantial. In Table S4 ‘Oxidative Phosphorylation’ is a subset of ‘Mitochondrial Dysfunction’; if at all a clear link to mitochondria, ‘iNOS signaling’ and ‘iNOS signaling in skeletal muscle’ contain the same genes. ‘Antioxidant Action of Vitamin C’ links as mitochondria are commonly considered as a major source of superoxide production. The Seahorse data on mitochondrial respiration have some technical and statistical shortcomings.
In general, it would be appropriate to differentiate gene expression in transcript level and protein level everywhere in the text and the figure legends.
The 3d graph is still not showing the 3rd dimension in an appreciable way.
VMSC housekeeping correction: using this approach would in my opinion just enhance differences that stem from variation in impurities, i.e. contribution from other cell types. As the ‘correction’ does not change much, it is questionable whether the original output or the corrected should be used. How would the data in Fig. 2A look without this correction?
‘..it is difficult to give quantitative measures to proteins analyzed by proteomics..’ I guess many proteomics specialists will disagree to this statement in the response to my criticism of the absent/present only evaluation. Using protein samples extracted from formalin-fixed paraffin embedded aorta tissue samples may be challenging. But there are multiple options to normalize and compare patterns. The absent/present criteria have the problem of defining the threshold for present. If the absent criterium was below threshold in all samples of one type, what were the present criteria: present in all samples of the respective type?
Seahorse: the tracings for control 1 and control 2 in Figure 5A and 5B look identical. Is this the average of 2 biological replicates with 5 wells each? It is not correct to consider the day to day and within day variation together (n=10).
I am no statistics specialist, but p-values <0.0001 in Figure 5C and 5D using Student’s t-test appear too high.
Round 3
Reviewer 1 Report
In consideration of the potential interest to readers, I am recommending paper be accepted despite the limitations previously noted.
Author Response
We would like to thank the reviewer for the time and effort necessary to review our manuscript.
Reviewer 3 Report
Issues are satisfactorily met.
Author Response

(The authors gave the same response as above.)
